# Extended gate field-effect-transistor for sensing cortisol stress hormone

Shokoofeh Sheibani[1,2], Luca Capua[1], Sadegh Kamaei[1], Sayedeh Shirin Afyouni Akbari[3], Junrui Zhang[4], Hoel Guerin[4] & Adrian M. Ionescu [1✉]

Cortisol is a hormone released in response to stress and is a major glucocorticoid produced by adrenal glands. Here, we report a wearable sensory electronic chip using label-free detection, based on a platinum/graphene aptamer extended gate field effect transistor (EG-FET) for the recognition of cortisol in biological buffers within the Debye screening length. The device shows promising experimental features for real-time monitoring of the circadian rhythm of cortisol in human sweat. We report a hysteresis-free EG-FET with a voltage sensitivity of the order of 14 mV/decade and current sensitivity up to 80% over the four decades of cortisol concentration. The detection limit is 0.2 nM over a wide range, between 1 nM and 10 μM, of cortisol concentrations in physiological fluid, with negligible drift over time and high selectivity. The dynamic range fully covers those in human sweat. We propose a comprehensive analysis and a unified, predictive analytical mapping of current sensitivity in all regimes of operation.

[1] Nanolab, Ecole Polytechnique Fédérale de Lausanne, Lausanne, Switzerland. [2] Center of Excellence in Electrochemistry, School of Chemistry, University of Tehran, Tehran, Iran. [3] Advanced NEMS Laboratory, Ecole Polytechnique Fédérale de Lausanne, Lausanne, Switzerland. [4] Xsensio SA, Lausanne, Switzerland. ✉email: adrian.ionescu@epfl.ch

There is strong evidence that chronic stress has a negative impact on human health[1]. Disorders such as obesity, metabolic syndrome, type two diabetes, heart diseases, allergy, anxiety, depression, fatigue syndrome, and burnout are often associated with dysfunctions of the stress axes[2,3]. Under psychological and/or emotional stress, the adrenal gland cortex of the kidney releases cortisol into the bloodstream. Cortisol is known as the "stress hormone"[4]. The important relationships existing between the stressors and cortisol release on major body functions concern the following: (i) immune functions, (ii) metabolism, (iii) neurochemistry, and (iv) cardiovascular functions[3]. Disturbances in cortisol secretion are thought to be a prime mediator in associations between stress and health. Cortisol has an important role in regulating carbohydrate metabolism to make the human body resist against pain and external stimuli, such as infection or psychological tension[5]. It has the ability to maintain homeostasis in the cardiovascular, immune, renal, skeletal, and endocrine systems[6]. The level of the cortisol has a circadian rhythm in serum throughout the whole day, with the highest level in the morning (~30 min after waking, 0.14–0.69 μM) and the lowest level at night (0.083–0.36 μM). Sustained stress can disrupt this rhythm and results in an abnormal increase of cortisol level[7]. Although the short-term activation of the hypothalamic–pituitary–adrenal axis is adaptive and necessary for everyday life, both high and low levels of cortisol, as well as disrupted circadian rhythms, are implicated in physical and psychological disorders.

Recently, there has been an increasing interest in sensing cortisol biomarker in biofluids for numerous diseases related to the stress. As the secreted cortisol enters into the circulatory system, it can be found in detectable quantities in several biofluids in human body including saliva, sweat and urine[8]. The cortisol circadian rhythm (Fig. 1a) and its variations and pulsatility at lower timescale may indicate the individual's specific acute reactivity to stressful situations[9,10]; there now exists a high demand for a sensing system capable to support daily quasi-continuous measurements[11]. In this context, the work presented here proposes and experimentally validates a sensitive and selective method for the quasi-continuous monitoring of cortisol levels in biofluids, being suitable for sweat analysis with a patch (Fig. 1b).

The traditional methods for detection of cortisol include immunoassays such as radio-immunoassay[12], enzyme-linked immunosorbent assay[13], and fluoroimmunoassay[14,15]. The detection limits for these methods are 1.59, 0.036, and 0.52 nM, respectively. In addition, normal and competitive immunosensors in conjunction with surface plasmon resonance transducer with the detection limit of 0.03 nM in urine and 0.13 nM in saliva have been reported[16,17]. The physiological values of cortisol levels in human perspiration range from about 1.4 nM to 0.4 μM[6]. Traditional detection methods are time consuming and complex, needing multiple-step reactions and washing processes for subsequent analyses. In addition, immunoassays require both labeled analogs of cortisol and elaborated hardware.

In the quest for achieving portable systems, the electrochemical sensors appear as an attractive alternative solution. Their main challenge is that the cortisol does not have a redox center. Inkjet-printed electrochemical biosensors have been proposed by exploiting metalloporphyrin-based macrocyclic catalyst ink for the direct reduction of the cortisol captured by the aptamers-functionalized magnetic nanoparticles. However, the preparation of magnetic nanoparticles and catalyst inks are not yet fully mature for high-performance commercial sensors[18]. Other electrochemical methods are based on electrochemical impedance spectroscopy (EIS) or cyclic voltammetry via a mediator like $Fe^{3+}/Fe^{4+}$. In these studies, the detection of the cortisol is achieved by the inhibition of the surface for redox reaction[19–23]. In addition, chemiresistor-immunosensors followed a similar strategy[24,25]. Among the electrochemical sensors, the ones using graphene and its derivatives[26] have shown very low

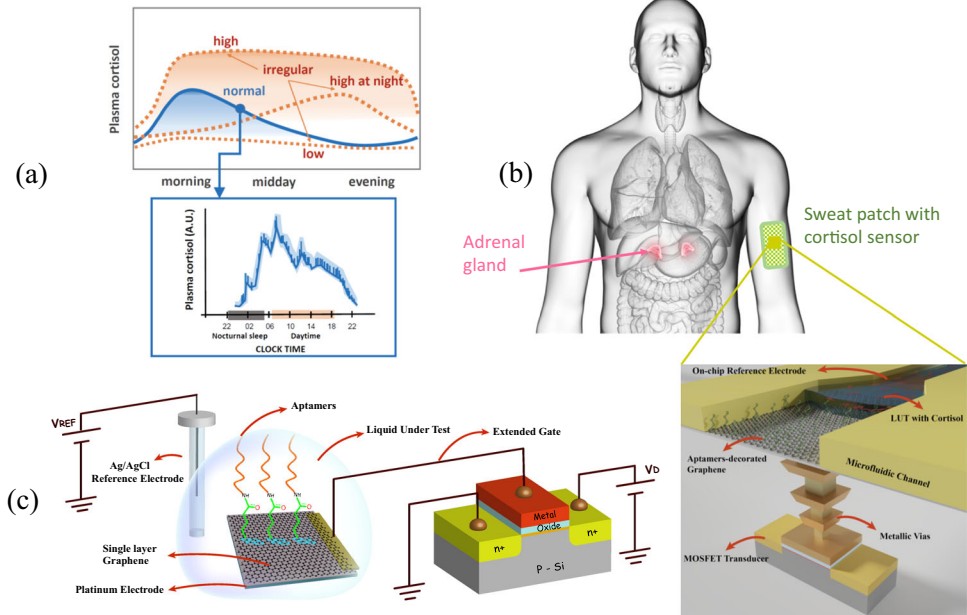

**Fig. 1 Cortisol circadian rhythm and sensing principle of the extended-gate FET configuration for its measurement in a wearable patch. a** Qualitative depiction of regular and irregular circadian levels of the cortisol produced by the adrenal glands in human body through the day, showing the need for high time granularity measurements to capture the pulsatile nature of cortisol, **b** concept of 3D-integrated cortisol sensor embedded in a patch used as wearable on the upper arm: at the bottom, the FET gate is extended through metal vias to a top metal layer covered with aptamer-decorated graphene. The system includes a top microfluidic channel that guides the sweat (LUT) over an integrated planar reference electrode and the functionalized graphene. **c** Schematic of extended-gate FET configuration used in this work: the functionalized electrode is electrically connected to the gate of the MOSFET transducer and immersed in the measurement solution while the extended-gate electrode is biased via the voltage applied on the reference electrode.

detection limit and high sensitivity, thanks to the conductivity and unique structural properties[21,25,26]. Graphene has the ability to attract a wide range of aromatic molecules, due to the $\pi$–$\pi$ interactions or electrostatic interactions and can be used to functionalize electrodes with bio-recognition probes such as antibodies or aptamers[26]. However, electrochemical methods still suffer of significant loss of selectivity because of nonspecific physical adsorption of other species existing in the medium and consequently and their scaling is quite limited.

In recent years, ion-sensitive field-effect transistors (ISFETs) have attracted a lot of attention due to their fast response, sensitivity, low power operation, ability to offer co-integrated readouts and full on chip-circuit design, miniaturization, and low cost[27–30]. All these features make them one of the most promising candidates for wearable systems. ISFETs form a subset of potentiometric sensors that are not affected by signal distortions arising from the environment, thanks to the input gate potential that is connected to the electrical FET transducer[31]. They are capable of converting any little variation of the electrical charge placed in the vicinity of the transistor gate, such as any species carrying charge (similarly to ions), become detectable by a variation of the FET drain current. The operation of an ISFET sensor is based on the dependence of the threshold voltage of a metal-oxide semiconductor field-effect transistor (MOSFET) on the gate work function, which can be modulated by the charge of an ion-sensitive membrane[31]. As state-of-the-art nano-MOSFETs operate at low voltage with low currents, ISFETs inherit their high charge sensitivity. Any chemical reactions at the top of the gate dielectric with the various species existing in the solution may induce a change of gate stack electrical characteristics. Therefore, the current–voltage characteristic of the ISFET sensor can be modulated if the gate dielectric is exposed to interactions with fluids. However, in an advanced complementary metal–oxide–semiconductor (CMOS) process, the gate stack is part of the so called front-end-of-line process that is highly standardized and cannot be easily modified or functionalized for sensing. To address this issue, extended-gate (EG) FETs have been proposed for sensing applications[32,33]. In such sensor architecture, the base transducer is a standard nano-MOSFET, whereas the sensing element is formed by a specific functional layer on the extension of the metal gate that can be an external electrode or a metal layer in the back-end-of-the-libe (BEOL), connected to the nano-MOSFET gate. The EG-FET configuration has major advantages due to the separation of the integrated transducing element from the functional layers, including higher stability, less drift and even less temperature sensitivity[34]. Few research groups have attempted the design of cortisol sensors exploiting FET devices, although they have faced some challenges and, to our best knowledge, we report in this work the first EG-FET that fulfills sensitivity and selectivity performance compatible with sensing in human biofluids.

One of the challenges of the FET-based sensors is the Debye screening effect in ionic liquids, which prevents its electrical potential to extend further than a certain distance, known as Debye length ($\lambda_D$). The value of $\lambda_D$ depends on the ionic strength of the liquid. For instance, $\lambda_D$ in phosphate-buffered saline (1× PBS), which is commonly used in biological research is <1 nm. The physical lengths of antibody–antigen complexes, usually utilized for ISFET biosensors, are larger than $\lambda_D$ associated with physiological media[35]. Therefore, the challenge for designing a FET sensor for detection of the cortisol is the choice of an appropriate catch probe overcoming the Debye length. As cortisol is charge-neutral, the electrical recognition of the cortisol is subject to the use of an electrically active mediator catch probes that have their own charge to modulate the gate potential within the detectable Debye length. Thus, the binding between the catch probe and the

cortisol will cause a change in the total gate potential and, consequently, in the measured drain current. Until now, different capturing probes including molecularly sensitive polymers, antibodies, and aptamers have been used in the reported ISFET devices for detection of the cortisol[36–38]. The use of aptamers, which is the solution adopted in in this work, has some clear attractive advantages. Aptamers are single-stranded nucleic acid molecules, which are negatively charged due to the presence of a phosphate group in each nucleotide of the nucleic acid strand. Aptamers can fold into three-dimensional (3D) topologies, with specifically designed pockets for binding with a target of interest[39]. Compared to antibodies, aptamers have superior advantages as catch probes as they are synthesized in vitro, reducing the batch-to-batch difference. In addition, they can be designed for different degrees of affinity for a targeted molecule vs. a modified disturbing analog[40,41]. Moreover, aptamers are less affected by temperature fluctuations and are more stable for long term storage[18,42]. They can be covalently immobilized on most surfaces by modifying the 5′- or 3′-end[43]. Three different apatmers have been introduced for detection of the cortisol. They have 40[44], 61[18], and 85[38,45] nucleotides. The one with 85 nucleotides has been previously applied for a FET sensor with the detection limit of 50 nM[38]. However, for a FET sensor facing the challenge of the Debye length, the shorter length of the aptamer is expected to have better sensitivity and lower detection limit as it has higher chance to not exceed the Debye length when it reacts with the target.

In this study, we demonstrate a label-free cortisol detection method with an EG-FETs, which is overcoming the Debye screening limitation for charge sensing by using by using 61-basepair aptamer-decorated single-layer graphene on platinum as a gate electrode. The sensing element is physically separated from the electrical transducer, enabling the possibility to implement the sensor in a 3D configuration, with a nano-MOSFET as base voltametric transducer, and the sensing electrode in the BEOL of a CMOS process, resulting in a low power wearable sensory electronic chip. The use of atomically thin graphene is crucial to chemically bind the aptamers and bring the recognition event of the analytes within the Debye screening length, with high sensitivity. The resultant limit of detection (LOD) is 0.2 nM. The reported EG-FET sensor is hysteresis-free and shows excellent selectivity towards cortisol in presence of other similar hormones. Further, it has a voltage sensitivity over the four decades of cortisol concentration, which fully covers the one in human sweat. We propose a, predictive analytical mapping of current sensitivity in all regimes of operation. To the best of our knowledge, this analytical model demonstrates for the first time a fully self-closed analytical dependence of sensor output current on the cortisol concentration over the whole range of concentrations in human sweat.

## Results and discussion

**Sensor configuration**. The EG-FET configuration proposed and explored in this work is depicted in Fig. 1c: the gate is extended through an external electrically connected platinum electrode, which is properly functionalized with a monolayer graphene sheet, decorated with selected aptamers to address the detection of the cortisol within the Debye length. The platinum/graphene electrode is then immersed in the measurement solution, where a standard Ag/AgCl reference electrode is used to electrically bias the gate of the MOSFET through the solution. Experiments have been carried out with a reference buffer and solutions with various known cortisol concentrations. Following cortisol catching, the resulting changes in the MOSFET drain current is recorded and analyzed. The significant advantage of this proposed configuration relies in using an extremely stable and reproducible standard 0.18 μm CMOS technology node for the FET transistor as transducer, while separately functionalizing the extended electrode.

This configuration results in the possibility to obtain a fully 3D lab-on-chip sensory system (Fig. 1b bottom) with the active detecting element in the BEOL, the specific Pt/graphene/aptamer layer, and the reference electrode (e.g., another fluorinated metal/graphene electrode) in the BEOL of the CMOS process. Such a 3D chip is compatible with the recently proposed concept of lab-on-skin (LOS)[46] suitable for collection and analyzing the concentrations of biomarkers in human sweat. Such conceptual LOS system (Fig. 1c right) includes an SU-8 integrated microfluidic system that allows the liquid under test to flow over both the planar chlorinated reference electrode and the graphene/aptamer sheet.

The transfer process of graphene onto the working electrode is described in the detailed experimental data in Supplementary Note 3. First, a thin layer of poly(methy methacrylate) is spin-coated onto the as grown graphene film on Cu substrate followed by a baking process and the etching of the growth substrate by ammonium persulfate 0.1 M solution. The floated polymer/graphene is rinsed with deionized (DI) water several times and then fished out on to the Pt electrode (please see its configuration in Supplementary Note 2) acting as an extended gate. In a last step, the polymer is dissolved with acetone and rinsing with isopropanol is applied. After the transfer process, the prepared electrode is functionalized with aptamers. Figure 2d, e show the scanning electron microscope (SEM) images of the transferred graphene on platinum before and after aptamers functionalization, confirming the uniformity and cleanness of the transferred graphene onto the Pt electrode and of the functionalization of graphene with aptamers. Moreover, the SEMs confirm that there are no defects and cracks on the graphene after the functionalization process; the visible dark lines in Figs. 2d, e indicate the boundary of the large graphene grains resulting from its fabrication process. The darker parts in the image after the functionalization are attributed to aptamers immobilization.

**Functionalization and characterization of the graphene electrode surface**. The chemistry and the different steps for the electrode modification as well as the following attachment of the targets are shown in Fig. 2a. To efficiently functionalize the electrode, 1-pyrenebutyric acid N-hydroxysuccinimide ester (PBSE) with the thickness of 0.75 nm is used as the linker molecule between the aptamers and the graphene sheet. PBSE can attach to the graphene surface by its carbon rings via $\pi$–$\pi$ interactions. Moreover, the length of the amine group, which is added at the 5′-end of the aptamer to enable it for the EDC (1-ethyl-3-(dimethylaminopropyl) carbodiimide) -NHS (N-hydroxysuccinimide) reaction with PBSE, is 0.91 nm. Therefore, the total distance between the aptamer and the graphene surface is 1.66 nm (Fig. 2b). Considering the fact that the Debye length in physiological salt environment (1× PBS), diluted 0.1× PBS, and 0.01× PBS are near 0.7, 2.4, and 7.4 nm, respectively[47], the Debye length in 0.05× PBS, which we used as the solution for doing the measurement of the response of the sensor, should be between 2.4 and 7.4 nm. Therefore, this method allows us to retain catch probe aptamers close to the conductive surface and prevent to exceed $\lambda_D$ and aptamers can induce their negative charges to the extended-gate electrode surface. The final modified electrode is electrically connected to the gate of long n-channel MOSFET fabricated in a 0.18 μm CMOS process.

To carefully evaluate the effect of the functionalization steps, a systematic series of experimental investigations based on EIS measurements, atomic force miscopy (AFM), and X-ray photoelectron spectroscopy (XPS) have been carried out on the fabricated structures after each step (see experimental details of these methods in Supplementary Note 4, 5, and 6). Figure 2c shows the EIS results in the form of Nyquist plots for the platinum-graphene electrode before and after the adsorption of

PBSE, and after subsequent immobilization of the aptamers on PBSE. The measurements were done in 0.05× PBS buffer solution containing 5 mM $Fe^{3+}/Fe^{4+}$. The charge transfer resistance, $R_{ct}$, is extracted from the Nyquist plot as the diameter of a semicircle observed in the high frequency range. This parameter describes the resistive behavior of the conjugated platinum-graphene electrode in contact with the electrolyte solution. In general, $R_{ct}$ values indicate a degree of surface coverage and hindrance of charge transfer at the electrolyte/electrode interface. For our electrode, the $R_{ct}$ value of the bare conjugated platinum-graphene electrode has been estimated to be 21 kΩ. This value increases to 31 kΩ after adsorption of the PBSE on the graphene surface. Moreover, a value of 39 kΩ has been measured after the immobilization of the aptamers. The increase of the $R_{ct}$ clearly indicates that the surface coverage increases and results in higher hindrance of the surface at each step of the electrode surface functionalization.

The characterizations of functionalized extended-gate electrode surface by AFM and XPS are summarized in Fig. 3. Figure 3a–c show the AFM images of the proposed electrode, after each step of the functionalization. As can be seen, after the adsorption of the PBSE and the immobilization of the aptamers, the roughness of the electrode surface is decreasing, which is in good agreement with other similar reports using aptamers for the electrode functionalization[48]. The formation of a complete layer of the PBSE and the immobilization of a monolayer of the aptamers on top of the graphene cause the significant decrease in the roughness of the electrode. Before the PBSE adsorption, the graphene surface shows an average roughness value of 59 nm, which decreases to 39 nm after the adsorption of the PBSE on top of the graphene. Furthermore, decrease of the roughness down to around 29 nm after the attachment of the aptamer is observed. In addition, XPS was performed to further prove the structural integrity of gate functionalization.

Figure 3d–f provide the results of the XPS analysis for the bare electrode with monolayer graphene sheet and treated graphene samples. The characteristic peak of N1s at 401.6 eV correspond to the presence of the NHS-ester ligands of the PBSE, which is in full accordance with previous reports[49]. The 0.8 eV shift in binding energy of the 401.6 eV peak after aptamers immobilization is attributed to the thymine homo-oligonucleotides used as simulated probes. In addition, the peak of N1s located at 399.3 eV could be assigned to the fraction of the probes being in contact with the substrate.

To further confirm the correct immobilization of the aptamers on the graphene surface, the transfer characteristic of the n-channel MOSFET was recorded after immobilization of the aptamers and the exposure of the full functionalized electrode to the cortisol (see detailed experimental data and results reported in Supplementary Note 8 and Supplementary Fig. 2). Once the abovementioned electrode functionalization has been completed, the EG-FET has been used to detect different cortisol concentrations in physiological buffer (PBS buffer 0.05×).

**Cortisol measurement**. The working mechanism of the proposed sensor and its figures of merit are described as follows. Our charge detection hypothesis is that the negatively charged aptamers approach the conductive electrode surface within the Debye length, due to the folding phenomenon, which arise from the binding of the cortisol to the aptamers. This binding event causes the strands to fold on themselves, and come closer to the surface[38]. Consequently, the surface potential of the electrode, ψ, is modulated by the cortisol concentration in solution. Due to the relation existing between the threshold voltage, $V_T$, and the surface potential at the interface between the electrolyte and the

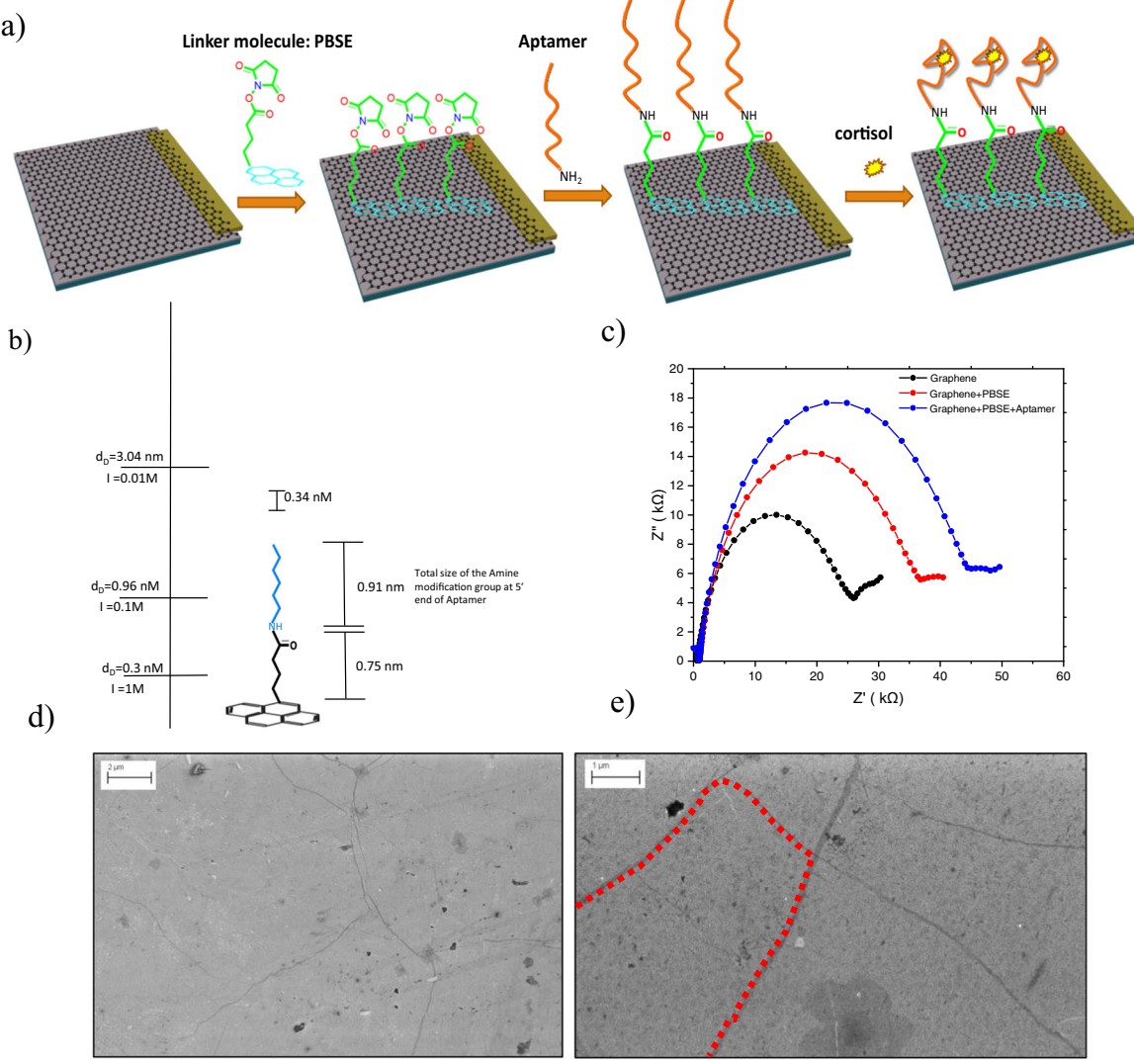

**Fig. 2 Functionalization steps of graphene electrode with aptamers and related experimental impedance and imaging. a** Schematic of the process flow with successive functionalization steps of graphene electrode. **b** Depiction of the estimated nanoscale thicknesses of the functionalization layers. **c** EIS curves at different functionalization steps. **d** SEM image of the bare transferred graphene on top of the Pt electrode. **e** SEM image of the transferred graphene after functionalization with aptamers. As explained in the Supplementary Note 1, the aptamer sequence used in this work is 5′-amine-AG CAG CAC AGA GGT CAGATG CAA ACC ACA CCT GAG TGG TTAGCG TAT GTC ATT TAC GGACC-3′.

sensing film[50], ψ, any change in the cortisol concentration, C, induces a change in the $V_T$ of the EG-FET detector:

$$V_{T\,(EGFET)} = V_{T\,(FET)} - \frac{\phi_M}{q} + E_{REF} + \chi^{Sol} - \psi(C) \qquad (1)$$

where the $V_{TFET}$ is the threshold voltage of MOSFET, $\phi_M$ is the work function of the metal gate relative to the vacuum, $E_{REF}$ is the potential of the reference electrode, and $\chi^{sol}$ is the surface dipole potential of the buffer solution. Therefore, at a voltage applied to the external gate, the surface potential, ψ, is modified by the amount of negative charges induced by the folded aptamers, which results in a right shift of the $I_{DS}-V_{REF}$ curves of n-channel MOSFET.

It is worth noting that the electrical dipole $\chi^{sol}$ at the interface between the metal gate and the electrolyte and the potential across the electrochemical double layer are the two phenomena that modulate the gate potential across the MOS. The value of $\chi^{sol}$ is influenced by different microscopic phenomena such as the distribution of charges in the immobilized chemical species, and the ionic physisorption and chemisorption exchange between the

modified gate and the electrolyte. As a result, the threshold voltage can be affected and hence deteriorate the sensitivity of an EG-FET[51]. In addition, the sensitive recognition of small molecules at low concentrations using the FET sensors may have particular challenges related to screening and size effects. Sensitive detection of small molecules at low concentrations via carbon-nanotube or graphene-based FET method is challenging due to the reduced electric field-effect of small size and few charge analyte and is even more difficult for uncharged analytes[52].

To validate the operation of the proposed device architecture for cortisol sensing and to extract its sensitivity, the sensor response to different cortisol concentration in buffer solution has been experimentally investigated (see details of the experimental set up in Supplementary Note 7). For this purpose, the transfer characteristics, $I_D-V_{GS}$, of the EG-FET transducer at different cortisol concentration in prepared buffer solutions, ranging between 1 nM and 10 μM (corresponding to cortisol concentrations in human biofluids such as plasma and sweat), have been systematically recorded at low drain voltage (100 mV), ensuring linear region operation. The goal is to achieve a high sensitivity in

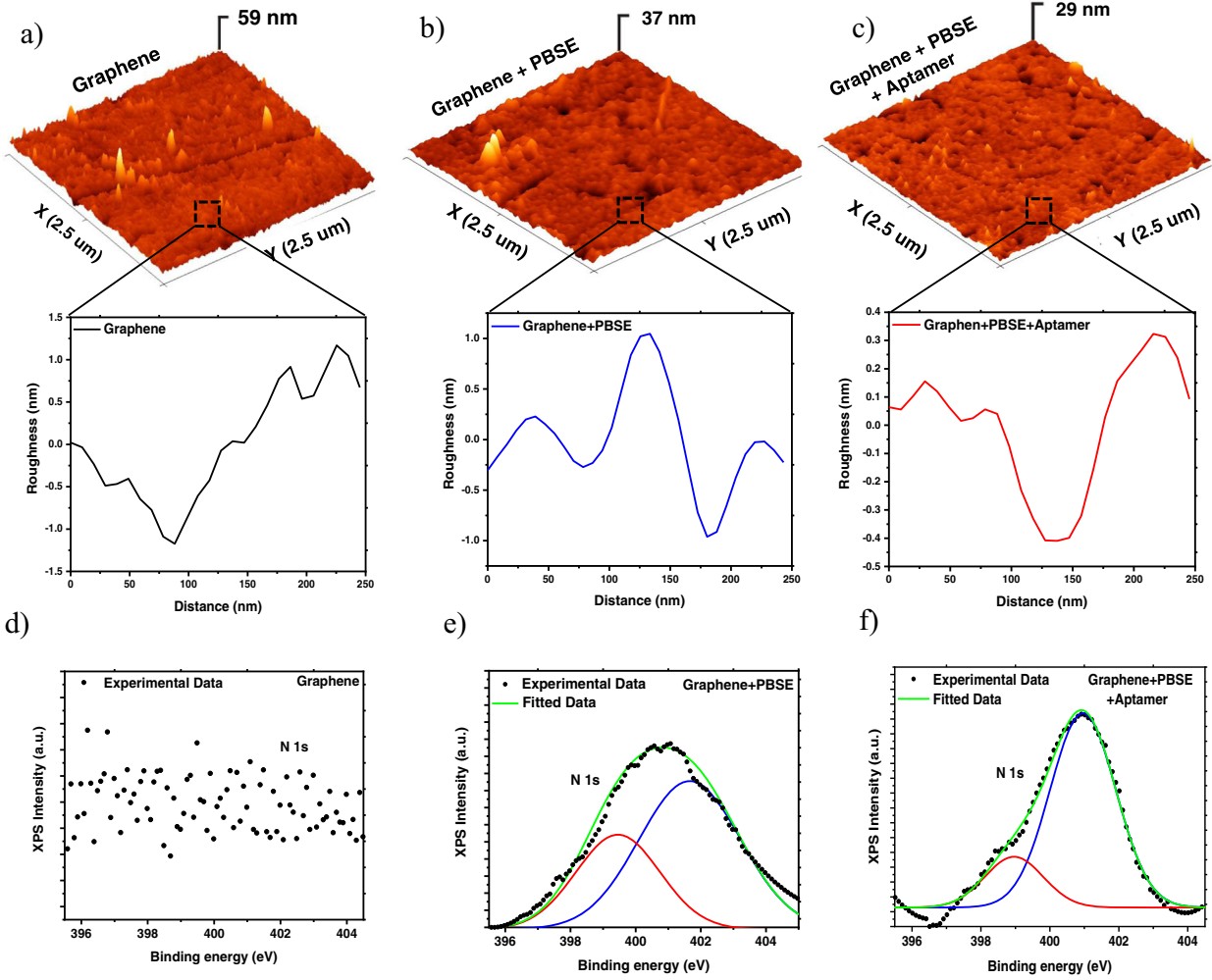

**Fig. 3 AFM 3D morphology of the Pt electrode and XPS characterization. a** After the graphene transfer, **b** after the PBSE linking, **c** after the aptamers functionalization. **d–f** Corresponding XPS graphs and fitting for different functionalization of the Pt electrode.

the whole range of cortisol concentrations (over four orders of magnitude) with a lower limit in the nM range. Therefore, the response of the EG-FET sensor has been studied in different regimes of inversion channel charge: (i) the weak inversion region (where voltage of the reference electrode ($V_{REF}$) is smaller than $V_T$ and the current is given by a diffusion mechanism) and (ii) the strong inversion region of operation (when $V_{REF}$ is larger than $V_T$ and the current is given by a drift mechanism).

It is well established that the modulation of the conductance of the FET-based sensors upon the binding of the target is correlated to the concentration when the gate and drain voltages are fixed. Figure 4a shows that after incubation of the different cortisol concentrations, $I_{DS}$–$V_{REF}$ curves shift to the right direction as the cortisol concentration increases. A notable achievement of our sensor functionalization is that $I_{DS}$–$V_{REF}$ curves show a negligible hysteresis, typically lower than 4 mV (Fig. 4b), and a small variation between repeated measurement with the same cortisol concentration. The extraction of the voltage shifts is performed at constant current both in subthreshold operation regime ($V_{REF} < V_T$) and in the strong inversion operation regime ($V_{REF} > V_T$) within a wide range of cortisol concentration, from 1 nM and 10 μM. In the Supplementary Note 9 and Fig. S3, we report cortisol sensor characteristics data with higher resolution in the low concentration range, between 1 and 100 nM, confirming the sensor response trend and the extracted sensitivity.

Two types of sensitivities are extracted to evaluate the figures of merit of the sensor: (i) a voltage sensitivity, $S_V = \frac{dV_{REF}}{d\log_{10}(\text{Conc})}\big|_{I_D=\text{const}}$, corresponding to the variation of the applied reference voltage to obtain the same drain current for different cortisol concentration, and, (ii) a current sensitivity, $S_I = 100\, x\, \frac{\Delta I}{I_0} = 100\, x\, \frac{|I_i - I_0|}{I_0}$, where $I_i$ is the current value at fixed gate voltage for a given concentration and $I_0$ is the current at a baseline lower concentration.

In the subthreshold regime, $S_V$ ranges between 11.9 and 14.7 mV/decade for different constant drain current levels, with the higher value measured for $I_{DS} = 1$ nA, while in strong inversion, it varies from 12.4 to 14.0 mV/decade. Our FET sensor shows similar voltage sensitivity for both working regimes, with a stable $S_V$ and excellent linearity (Fig. 4c–d) for detecting cortisol over 4 decades of concentration, demonstrating the full sensing capability of the designed aptamer-based catch mechanism. The LOD of the sensor is 0.2 nM. The amount of LOD is dependent on the sensitivity of the sensor. As previously discussed, the sensitivity is limited by the additional phenomena affecting the $\chi^{sol}$. Moreover, it is reported that the surface of the graphene has a tendency to attract some biological molecules[53]. Therefore, a high concentration of the aptamer (100 μM) is used for the functionalization of the electrode to cover the surface of the graphene by aptamers as densely as possible and minimize the free graphene spaces, and therefore decrease any unspecific

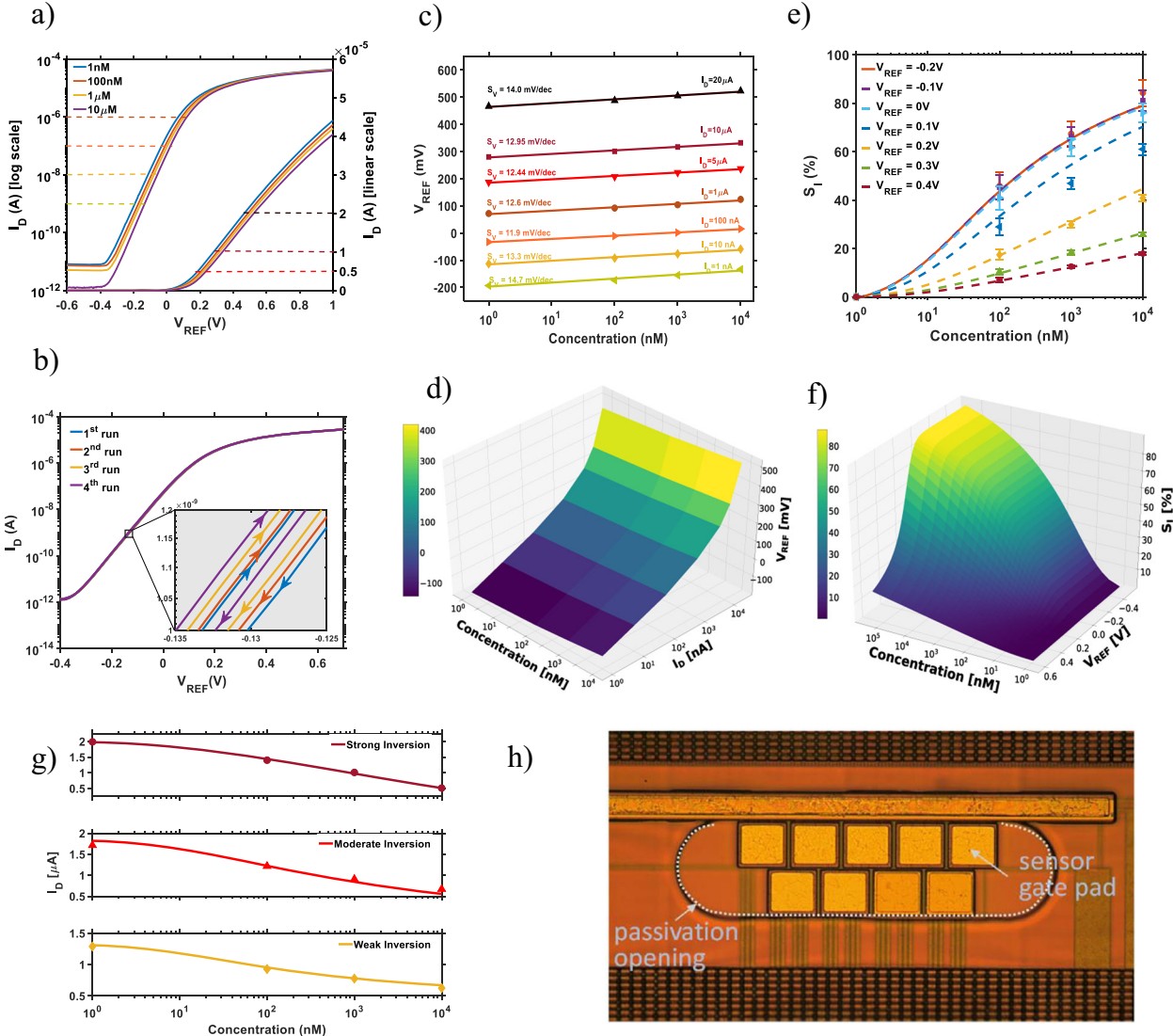

**Fig. 4 Experimental electrical validation of the proposed cortisol sensor and extracted sensitivities. a** $I_D$–$V_G$ transfer characteristics of cortisol sensor, in semi-logarithmic and linear scale for different cortisol concentrations. **b** Drift study and hysteresis. **c** Voltage sensitivity, $S_V$, for different current levels in subthreshold and in linear working regime. **d** Experimental 3D mapping of extracted voltage sensitivity as a function of concentration and drain current (all range of experiments). **e** Comparison between the current sensitivity in the two working regimes, the solid symbols indicate the real data and the dashed lines indicate the expected current sensitivity based on calibrated modeling. **f** 3D mapping current sensitivity, $S_I$, for different applied voltages (all operation range) and analyte concentration based on calibrated compact modeling using unified model of Eq. (2). **g** Predictive drain current model (solid line), based on Eq. (2), vs. cortisol concentration compared to data points (symbols) in weak ($V_{REF} = -0.2$ V), moderate ($V_{REF} = 0.1$ V) and strong inversion ($V_{REF} = 0.4$ V). **h** Photo of the 0.18μm CMOS silicon chip designed and used in this work; the central pads are the top gate contacts of FET sensors.

attachment of the molecules on the surface of the graphene. It should be noted that a too dense population of the aptamers on the surface of the graphene may restrict them to bend freely after attachment to the cortisol as the result of space disturbance by the neighbor aptamers. This phenomenon creates a trade-off and limits the sensitivity of this sensor and the corresponding LOD.

A noticeable difference in the performances of the sensor in the two regimes is obtained for $S_I$, due to the exponential dependence between the subthreshold drain current and the threshold voltage in weak inversion, compared to the strong inversion where the current depends quasi-linearly on the threshold voltage. This difference is clearly shown in Fig. 4e, where $S_I$ has been evaluated for the two different regions (with corresponding error bars). Although the relative current change reaches values near 80% for the highest cortisol concentration in the subthreshold regime, it is limited to about 20% in the strong inversion regime. Such

exponential dependence in the weak inversion regime plays an important role considering the relative current changes for different concentrations, opening the path to a higher sensor resolution in this regime.

To analyze the sensor response in all the working regimes of the FET, for the whole cortisol concentration range in human biofluids, a compact physical sensor model was developed. The drain current is modeled with the following unified equation that accurately describes, the weak, moderate and strong inversion regions[54,55] of a long channel MOSFET:

$$I_D(V_{GS}) = \eta U_T^2 K_n ln\left(e^{\frac{V_{GS}-V_T}{\eta U_T}} + 1\right)\left(1 - e^{\frac{-V_{DS}}{U_T}}\right) \quad (2)$$

where $\eta = \delta V_{GS}/\delta \psi_S$ is the transistor body factor ($= 1 + C_{ox}/C_{dep} > 1$), $U_T = kT/q$ is the thermal voltage, $K_n = \frac{W}{L}\mu_0 C_{ox}$, $W/L$ is the channel width over length ratio, $\mu_0$ is the

low-field mobility, and $C_{ox}$ and $C_{dep}$ are the gate oxide and depletion capacitance, respectively. The experimental $I_D$–$V_{GS}$ curves at a given cortisol concentration are excellently approximated over the whole range of operation (see Supplementary Note 10 and Figs. S4a–b) by this model. Equation (2) is uniquely adapted to investigate a FET sensor, as it captures the role of threshold voltage, body factor, and temperature in a single unified equation, which can be simplified into traditional equations per regimes of operation (see Supplementary Note 10), being mainly based on physical parameters. Figure 4e compares the discrete experimental values of the current sensitivity, $S_I$, with the unified current sensitivity model in all working regimes. Based on the calibrated Eq. (2) we have mapped, $S_I$, for the full range of applied voltages and analyte concentrations, as shown in Fig. 4f. This physics-based calibrated analysis is very useful for choosing the optimal sensor bias for the desired $S_I$, sensitivity, as well as for trade-off with the lowest power consumption or the lowest signal-to-noise ratio, according to the final sensor application. By combining Eq. (2) with the threshold voltage dependence on analyte concentration (see Supplementary Note 10, Eq. S3, and Fig. S4b), we derive a closed nonlinear logarithmic expression of the dependence of the FET sensor current, $I_{DS}$, on the cortisol concentration, $C$, for every bias point:

$$I_D(V_{GS}) = \eta U_T^2 K_n ln\left(1 + \left(\frac{C}{C_{ref}}\right)^{-\frac{m}{\eta U_T}} e^{\frac{V_{GS}-V_T}{\eta U_T}}\right)\left(1 - e^{\frac{-V_{DS}}{U_T}}\right) \quad (3)$$

where m is a non-ideality factor that characterize the sensor efficiency and could potentially capture specific Langmuir adsorption surface phenomena, whereas $C_{ref}$ is the lowest concentration (1 nM) investigated in the reported series of experiments of Fig. 4a, taken as a normalizing reference. The accuracy of Eq. (2), valid in all regions of operation of the sensor, is confirmed in Fig. 4g using the array of FETs of fabricated chip sensor of Fig. 4h. The accuracy of the full model is further confirmed in Supplementary Fig. S5. This is the first unified analytical expression capable to precisely predict FET sensor response to the cortisol, to analytically capture the sensing performance and optimize the signal-to-noise ratio and power consumption.

Finally, two other important figures of merit of the proposed cortisol sensor have been studied and reported here: (i) the sensor selectivity, which describes the specificity of the sensor towards the target in the presence of interfering compound, and (ii) the drift of the response caused by the environmental effects over time. They are both crucial for designing an accurate sensor and for employing it to produce high quality reliable data in practice. The selectivity of the sensor to other types of human hormone and the drift effect is reported for the fabricated sensor in Supplementary Note 11 and 12. To study the selectivity, we investigated the effect of the testosterone hormone, another adrenal hormone with similar structure to the cortisol, and cortisone, metabolized form of cortisol in the peripheral tissue. The proposed sensor was exposed to the different controlled concentrations of the testosterone in the range of human biofluids and cortisone in the range of the concentrations similar to the cortisol measurement. Then, the transfer characteristics of the EG-FET have been recorded and reported in the Supplementary Fig. 6. No significant trend is observed for $I_{DS}$-$V_{REF}$ curves as the testosterone or cortisone concentration increases, which validates the high selectivity of our aptamer functionalization. In addition, the drift in the response of the sensor was investigated by immersing the sensor into an incubation buffer for 30 min for three consecutive times and by recording the sensor response. The procedure and the results are depicted in the Supplementary Note 12 and Fig. S7, revealing that no trend significant in the $I_{DS}$–$V_{REF}$ was observed after 1.5 h, which demonstrates that the

proposed cortisol sensor based on aptamer functionalization has a very stable, drift-independent response.

## Conclusion

We proposed and demonstrated a successful design for an EG-FET sensor for selective recognition of the cortisol hormone by the exploitation of single-layer graphene and aptamers as the gate electrode and catch probes, respectively. The utilization of the aptamers as the recognition elements make our sensor highly sensitive, selective and stable. Our EG-FET is hysteresis-free and showed unique sub-nanomolar detection limit, negligible drift and high selectivity over a wide dynamic range of concentration. Its dynamic range and low detection limit make it a promising candidate for the detection of normal and abnormal amount of the cortisol in biofluids such as sweat saliva and serum. A compact model for the drain current in all regimes of operations, useful for sensor optimized design, was proposed and validated. This enabled the derivation of the first analytical expression of the sensor output current as a function of the cortisol concentration with high predictive capability. These features make this sensor an excellent candidate for integrated miniaturized lab-on-chip or LOS wearable sensory systems capable to monitor the concentration of cortisol in human biofluids.

## Methods

**Electrode functionalization with aptamers.** The graphene surface was functionalized with PBSE by immersion of the chip in 10 mM solution of PBSE in dimethylformamide (DMF) for 2 h. The prepared chip was gently rinsed with DMF and DI water, and then dried with $N_2$ gas. The PBSE molecule contains a pyrene group that binds to graphene via $\pi$–$\pi$ interaction and an ester group at the other end, which reacts with primary amines. The immobilization of aptamers was done by placing 2 μl drop of 100 μL aptamer solution in DI (overnight, in wet chamber at 4 °C) on the graphene surface modified by the PBSE. The amine group at 5′ end of the aptamers reacts with the ester end of PBSE by NHS reaction.

**Cortisol sensing method with EG-FET sensor.** The stock solution of the hydrocortisone with the concentration of 0.01 M was made in methanol. Then all the other diluted solutions were prepared from the stock solution in the incubation buffer (50 mM Tris-HCl buffer containing 100 mM NaCl and 5 mM $MgCl_2$ pH 7.4). To investigate the sensor response under different cortisol concentration, the modified electrode was immersed in the incubation buffer with a known concentration of the cortisol for 30 min, followed by rinsing with DI water five times and drying with $N_2$ gas gently. Then, the electrode was placed in the static cell filled with PBS buffer (0.05×, pH 7.4) as the measurement solution. The $V_{REF}$ was swept between 0 and 2 V with the step of 40 mV, whereas a constant voltage of 0.1 V was applied between source and drain contacts to ensure that the MOSFET operates in the linear regime. The $I_{DS}$–$V_{REF}$ characteristics of the n-channel MOSFET were recorded as the response of the EG-FET sensor for different cortisol concentrations.

## Data availability

The data that support the findings of this study are available from the corresponding author upon reasonable request.

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

## Acknowledgements

The contributions of L.C. and S.K. have been funded by the Swiss National Science Foundation via the CONVERGENCE Flag Era Project and by the European Research Council (ERC) Advanced Grant "Milli-Tech" under Grant 695459, respectively.

## Author contributions

A.M.I., S.S., S.K., and L.C. co-wrote the paper. S.S and A.M.I. jointly made technological and experimental choices, with A.M.I, supervising interactions with the other authors. S.S. planned and performed most of the experiments and measurements. S.S. and A.A. contributed to transfer of the graphene on the electrode. S.K. did SEM imaging and analyzed the data of XPS and AFM experiments. L.C. contributed to the predictive analytical mode of the article. J.Z. and H.G. designed the PCB and the chip of the MOSFET, and helped with the *I–V* experimental characterization and the *I–V* interpretations. All authors have verified and approved the final version of the manuscript.

## Competing interests

The authors declare no competing interests.

## Additional information

**Peer review information** Primary handling editor: John Plummer.

