## [Peer Review File · Communications Materials]

Web links to the author's journal account have been redacted from the decision letters as indicated to maintain confidentiality.

22nd Jun 20

Dear Professor Ionescu,

Thank you for submitting your manuscript, "An Extended Gate Field-Effect-Transistor for Sensing the Hormone of Stress in Human Biofluids", to Communications Materials. It has now been seen by 3 referees, whose comments are appended below. You will see that while they find your work of potential interest, they have raised substantial concerns that must be addressed. In light of these comments, we cannot accept the manuscript for publication, but would be interested in considering a revised version that addresses these serious concerns.

In particular, you will see that Reviewer 1 raises a number of concerns regarding unsupported claims in the paper, requiring additional data to understand sensing performance, and is asking for more information regarding the sensing mechanism. Naturally, we expect these concerns to be convincingly addressed in the revised paper, as well as the comments from the other referees.

We hope you will find the referees' comments useful as you decide how to proceed. Should further experimental data or analysis allow you to address these criticisms, we would be happy to look at a substantially revised manuscript. However, please bear in mind that we will be reluctant to approach the referees again in the absence of major revisions. If the revision process takes significantly longer than three months, we will be happy to reconsider your paper at a later date, as long as nothing similar has been accepted for publication at Communications Materials or published elsewhere in the meantime.

When submitting your revised manuscript, please include the following:

-A rebuttal letter with a point-by-point response to each of the referee comments and a description of changes made. Please include the complete referee report in the rebuttal letter. Please note that the rebuttal letter must be separate to the cover letter to the editors.

-A marked-up version of the manuscript with all changes to the text in red colored font. Please do not include tracked changes or comments. Please select the file type 'Revised Manuscript - Marked Up' when uploading the manuscript file to our online system.

-A clean version of the manuscript. Please select the file type 'Article File'.

-An updated <https://www.nature.com/documents/nr-editorial-policy-checklist.zip> Editorial Policy checklist, uploaded as a 'Related Manuscript File' type. This checklist is to ensure your paper complies with all relevant editorial policies. If needed, please revise your manuscript in response to these points. Please note that this form is a dynamic 'smart pdf' and must therefore be downloaded and completed in Adobe Reader. Clicking this link will download a zip

file containing the pdf.

Please use the following link to submit your revised manuscript files:

[link redacted]

We understand that due to the current global situation, the time required for revision may be longer than usual. We would appreciate it if you could keep us informed about an estimated timescale for resubmission, to facilitate our planning. Of course, if you are unable to estimate, we are happy to accommodate necessary extensions nevertheless.

Please do not hesitate to contact me if you have any questions or would like to discuss the required revisions further. Thank you for the opportunity to review your work.

Best regards,

John Plummer, PhD
Chief Editor
orcid.org/0000-0003-4824-8497
Communications Materials

Reviewers' comments:

Reviewer #1 (Remarks to the Author):

The authors developed an extended gate field-effect transistor based on an aptamer decorated graphene-platinum electrode connected to the FET gate for the detection of cortisol in phosphate-buffered saline. The use of PBSE linkers with aptamer contains the receptor layer within the debye length upon cortisol binding, which subsequently modulates the surface potential. Despite the emphasis of biological fluids in the title, concept figures and substantial discussion contributed to the physiological and clinical relevance of cortisol; there was no study related to the implementation of the EGFET in a wearable or point-of-care format nor has it been applied for real human biofluids analysis. Unfortunately, I cannot recommend its publication in Communication Materials without major revision. The authors should back up their claims with more substantial studies.

While the sensors demonstrated a broad dynamic range for cortisol detection (1 nM -10 μ M) in buffer; its application in the lower detection range is not demonstrated or well explained. For a 14 mV/decade sensitivity, the detection of cortisol in sweat or saliva (actual human biofluids) in the order of several nanomolar requires millivolts precision for the accurate determination of cortisol. The authors only in effect demonstrated a clear separation of signals for 1 nM from 100 nM, which is

not physiologically relevant. The authors also claimed that the use of current sensitivity by normalizing the current change with regard to baseline level rendered the sensor “higher resolution” depending on the choice of voltage. Although relative current changes differ by 60% when operated under different regimes; the relative errors of sensor operating under different regimes were not presented. The authors did mention this trade-off with the analytical solution of drain current dependence on cortisol; however, this was not explained in details. A demonstration of how Equations (2) and (3) can be used to optimize sensor application and performance should be provided. The authors should leverage this study to prove whether the sensor can indeed be applied to a lower concentration regime before claiming that the sensor covers the physiologically relevant range. Proper calibration based on experimental data for 0.2-10 nM cortisol should be provided instead of supporting the author’s sensitivity claim with a single I_{ds} - V_{ref} curve for 0.2 nM.

Figure 1c schematic and p6 line154-171 suggest that the sensor operation is done by biasing the extended gate electrode while immersed in ‘liquid under test’; whereas the experimental procedures suggest that separate buffers were used for incubation and measurement. Is there a stringent requirement to the measurement solution? The authors went into details about the Debye screening length in the sensor design yet ignored this discussion in the sensor operation principle. The schematic also included the SU-8 integrated microfluidic system but no actual attempt was made towards this integration. Figure 1c seems to be a conceptual illustration rather than a schematic that explains the operation principle of the chip that is actually fabricated and presented in this work.

Another ambiguous point is the surface modification process. The PBSE linkers are not blocked after aptamer immobilization. There is no evidence to suggest that all NHS-esters are deactivated after aptamer immobilization. Although cortisol and testosterone do not contain amine for their direct attachment to the NHS ester; tris(hydroxymethyl)aminomethane in the incubation buffer (there is a typo in SI for tris buffer) will potentially bind to electrode due to unreacted ester. Similarly, any amine-containing molecules in LUT might bind to the surface if there is excess NHS ester present. Please clarify this point.

The authors should perform selectivity study with more structurally similar molecules, especially cortisone. In addition, the citation for the aptamer sequence should be provided.

Other small issues related to the manuscript are:

p2 line41: pain is a sensation not a stimulus;

Labeling and formatting of Figure 4 are different from other figures.

Reviewer #2 (Remarks to the Author):

- Authors have developed an extended-gate TFT sensor for the cortisol detection. There are many advantages with the EG and aptamer probe. However, there are several issues that should be clarified.

- 1. For aptamer probe, authors have used 61 basepair aptamer, which has a reference. However, there are other sequences for the cortisol aptamer, even more different lengths of the aptamer have been reported. There are papers needed to be discussed in terms of the aptamers. (1) Dalirirad, et al, *Sens. Actuat. B*, 283 (2019) 79-86, (2) Jo, et al, *Sens. Actuat. B*, 304 (2020) 127424.
- 2. In the discussion of Debye length issue, there is a paper (Nakatsuka, et al, *Science*, 6750 (2018)) about the detection of small molecule like the cortisol. They used around 40-mers to capture the small molecules like neuro-chemicals. What will be an appropriate length for cortisol specific binding?
- 3. There are more EG TFT sensors to be compared with this study. Introduction part has missed importance of EG's roles. Authors can have more informative introduction.
- 4. The level of cortisol detection was relatively high. LOD should be or could be lowered by more measurements. If the LOD cannot be improved, there should be some discussions about the reason, apart from the issues of human's cortisol levels.
- 5. There are no introduction about how the cortisol concentration of sweat and that of blood plasma serum can be correlated to be a meaningful tool as a wearable sensor.
- 6. The comment about the COVID-19 does not seem to be very closely tied with cortisol sensor, especially with sweat sensor, even though the cortisol is one of glucocorticoids.

Reviewer #3 (Remarks to the Author):

What are the major claims of the paper?

The authors claim this work to be the first report on a wearable electronic chip based sensor utilising an aptamer/graphene/platinum Extended Gate Field Effect transistor (EG-FET). The use of aptamers allowed the cortisol detection phenomena to take place within the Debye length of the FET, thus ensuring high sensitivity. The achieved LOD was 0.2nM and the selectivity of the system was tested using alternate molecules such as testosterone, which showed no response. Moreover, the authors claimed to have observed negligible voltage drift in the proposed system, preserving the sensitivity of the device.

Are they novel and will they be of interest to others in the community and the wider field?

The FET based detection of cortisol is a well-known phenomenon. However, developing selective systems is still a challenge. The authors have used an aptamer based approach in their proposed methodology, which already exists in literature but still lacks extensive study. The authors still need to highlight their advantages over some of the highly sensitive reports on similar systems (for ex. Sekar et al., *Scientific Reports*, 2019 or Jeong et al., *Biosensors and Bioelectronics*, 2019).

Is the work convincing, and if not, what further evidence would be required to strengthen the conclusions?

The work is convincing. The characterisation and the modelling has been extensive and satisfying in justifying the results. However, one small comment for the AFM study: The AFM trend needs to be explained. The SEM images seem to suggest that the roughness increases. Why is the trend different in AFM?

On a more subjective note, do you feel that the paper will influence thinking in the field?

Yes. As mentioned above, the authors explore using aptamers for a robust FET device. This area of research still being niche, is relevant and helpful for other researchers working in similar areas.

Please feel free to raise any further questions and concerns about the paper.

- The authors show that IDS-VREF curves show no significant shift at high concentrations but have a slight left shift at low concentrations. This observation has not been explained properly.
- On a general note, the authors claim the biggest advantage of the system is that the folding of aptamers helps the detection to take place within the Debye length, however Debye length is also dependent on the matrix used (in this case PBS). Does it mean that the use of aptamers could be made redundant if an appropriate matrix is used? Moreover, it would be interesting to see the device performance in other matrices as well.

We would also be grateful if you could comment on the appropriateness and validity of any statistical analysis, as well the ability of a researcher to reproduce the work, given the level of detail provided.

The paper is well written and the methods have been well documented. Therefore, other researchers can follow and reproduce the work without difficulty.

Reviewer 1

The authors developed an extended gate field-effect transistor based on an aptamer decorated graphene-platinum electrode connected to the FET gate for the detection of cortisol in phosphate-buffered saline. The use of PBSE linkers with aptamer contains the receptor layer within the Debye length upon cortisol binding, which subsequently modulates the surface potential

1. **Despite the emphasis of biological fluids in the title, concept figures and substantial discussion contributed to the physiological and clinical relevance of cortisol; there was no study related to the implementation of the EGFET in a wearable or point-of-care format nor has it been applied for real human biofluids analysis.**

Thank you for your constructive comments. We agree that we should clarify and improve some aspects of the article. For this purpose, as it can be observed in all the answers we carried out new experiments and we significantly improved the new version of the article. All the changes in the main paper are marked with **yellow**.

The article is focusing on the first steps of the developing of a new method for detection of cortisol. We studied the selectivity and the stability of this sensor for detection of cortisol in PBS physiological buffer, which is commonly accepted physiological buffer for the bio measurements, as the osmolality and ion concentrations of this solution usually match those of the human body. The study and the experimental validations have been carried out within cortisol concentration range in the physiological buffer that fully correspond to the application of the proposed sensor for sensing in sweat and interstitial fluid (ISF). Then we suggested that the configuration of the investigated sensory system results in obtaining a fully 3D lab-on-chip sensory system in combination with the chip design that we have created within our group, for a wearable sensor. However, we agree with reviewer comments about clarity and, following reviewer's remark, we removed the explicit keyword 'biological fluids' from the title, to avoid any confusion: *An Extended Gate Field-Effect-Transistor for Sensing the Hormone of Stress*.

2. **While the sensors demonstrated a broad dynamic range for cortisol detection (1 nM -10 μ M) in buffer; its application in the lower detection range is not demonstrated or well explained. For a 14 mV/decade sensitivity, the detection of cortisol in sweat or saliva (actual human biofluids) in the order of several**

nanomolar requires millivolts precision for the accurate determination of cortisol. The authors only in effect demonstrated a clear separation of signals for 1 nM from 100 nM, which is not physiologically relevant.

Following this important remark, we prepared a new series of sensors and we specifically carried out a new experiment for cortisol concentrations between 1 and 100 nM to confirm the ability of the sensor to detect the low concentrations of the cortisol with proper resolution. In **Fig. S3, which is now added to supplementary information**, the separation between the signals of the different concentration is clearly showed and, even the sensitivity of the sensor for these lower concentrations is a little higher (between 15.7 and 16.9mV/decade) yet very close to the initial report in the paper. This new experiment reinforces our confidence that **all the results are solidly reproducible and the sensitivity as well, are fully confirmed in the low concentration range.**

Fig. S3: a) I_D -(V_{REF} - V_T) transfer characteristics of cortisol sensor, in semi-logarithmic and linear scale for different cortisol concentrations. b) Voltage sensitivity, S_V , for different current levels in subthreshold and in linear working regime.

3. *The authors also claimed that the use of current sensitivity by normalizing the current change with regard to baseline level rendered the sensor “higher resolution” depending on the choice of voltage. Although relative current changes differ by 60% when operated under different regimes; the relative errors of sensor operating under different regimes were not presented. The authors did mention this trade-off with the analytical solution of drain current*

We were not very sure of the reviewer request here but **we interpreted as a need for an error propagation study** that has been performed, expressing the relative errors for each concentration and fixed voltage. This has been possible because the drain current has been carefully recorded four different times for each concentration (which the review would agree is not reported and confirmed in many other publications) with a delay of five minutes between each experiment. The error calculated from averaging these four measurements for each concentration has been propagated according to the following equation:

$$\delta S_I = S_I \cdot \sqrt{\left(\frac{\delta(I - I_0)}{(I - I_0)}\right)^2 + \left(\frac{\delta(I_0)}{I_0}\right)^2}$$

$$\delta(I - I_0) = \sqrt{(\delta I)^2 + (\delta I_0)^2}$$

As expected, the relative errors decreases in the strong inversion working region, while the sensitivity has the opposite behaviour. In addition, the transconductance efficiency for different cortisol concentration is provided. As expected, the highest values correspond to the subthreshold region¹ where the device is characterized by a better amplification to power consumption ratio. Equations (2) and (3) of the main paper (unified compact models) are very suited to be integrated with the abovementioned considerations, in order to effectively and quickly recognize which is the best gate voltage potential at which the user (a read-out circuit designer) should bias the sensor, considering a trade-off between the desired sensitivity, gain and noise level. **These calculated error bars are now added to Fig 4e in the main article.**

Fig. 1: a) S_I values calculated with the reported equation, and the fitting model (dashed lines) for different voltage levels in all working regimes. Error bars are obtained with error propagation method.

b) Trans-conductance efficiency for different cortisol concentration. The plot highlights that the highest values are obtained in the subthreshold region, as expected.

- 4. The authors should leverage this study to prove whether the sensor can indeed be applied to a lower concentration regime before claiming that the sensor covers the physiologically relevant range. Proper calibration based on experimental data for 0.2-10 nM cortisol should be provided instead of supporting the author's sensitivity claim with a single I_D - V_{REF} curve for 0.2 nM.**

The level of the cortisol in sweat, which is the desired biofluid for the future proposed application of this sensor, is between 20 nM and $0.5\mu\text{M}^2$. However, we did a new set of experiments with higher concentration granularity (new Fig S3) to confirm the ability of this sensor for application in such low concentrations of the cortisol with a proper resolution. The sensitivity of the sensor for the tested new low concentrations is close and even a little bit higher than what we reported before, on the initial set of data that we reported in the main paper over a larger range of concentrations. The calculation of the LOD on the base of the signal of the blank and using the formula of the new calibration curve yields the limit of quantification of 0.14 nM, which confirms the very close to this value low claimed concentration of 0.2 nM reported before and tested experimentally. However, even if calculated detection limit is less than 0.2 nM, we still prefer to report 0.2 nM for the detection limit, as we could successfully make a difference between the signal of this low concentration and interfering hormone (testosterone) according to pervious experiments (supplementary Note 4).

- 5. Figure 1c schematic and p6 line154-171 suggest that the sensor operation is done by biasing the extended gate electrode while immersed in 'liquid under test'; whereas the experimental procedures suggest that separate buffers were used for incubation and measurement. Is there a stringent requirement to the measurement solution? The authors went into details about the Debye screening length in the sensor design yet ignored this discussion in the sensor operation principle.**

This is correct, thank you for pointing out. Fig. 1c is the illustration of our practical work and we changed accordingly the word **measurement solution** instead of **liquid under test** in that figure to respect your observation. This point is edited in the new version.

We used two different buffer solutions: the incubation buffer solution and the measurement buffer solution. The incubation solution should be a buffer with high ionic strength making the

incubation of the cortisol to the aptamer as effective as possible³⁻⁵. We applied an incubation buffer approved by other groups for incubation of the cortisol to the aptamer^{3,5}. For the measurement of the target, we used diluted PBS buffer to enhance the Debye length⁶. The amount of Debye length for different concentration of PBS is illustrated in Fig. 2b and the effect of PBS concentration on Debye length and the operation of the sensor is discussed in the article as follows:

(lines 206-218) 'To efficiently functionalize the electrode, 1-Pyrenebutyric acid N-hydroxysuccinimide ester (PBSE) with the thickness of 0.75 nm is used as the linker molecule between the aptamers and the graphene sheet. PBSE can attach to the graphene surface by its carbon rings via $\pi-\pi$ interactions. Moreover, the length of the amine group, which is added at the 5' end of the aptamer to enable it for the EDS-NHS reaction with PBSE, is 0.91 nm. Therefore, the total distance between the aptamer and the graphene surface is 1.66 nm (Fig. 2b). Considering the fact that the Debye length in physiological salt environment (1X PBS), diluted 0.1X PBS and 0.01X PBS are near 0.7 nm, 2.4 and 7.4 nm, respectively⁷, the Debye length in 0.05X PBS, which we used as the solution for doing the measurement of the response of the sensor, should be between 2.4 and 7.4 nm. Therefore, this method allows us to retain catch probe aptamers close to the conductive surface and prevent to exceed λ_D and aptamers can induce their negative charges to the extended gate electrode surface.'

Also, this effect is supported by the shift of the I_D-V_{REF} curve to the right direction after introducing of the aptamers to the extended gate electrode surface (Fig. S2). When the aptamers are retained within Debye length, the sign of folding them after attachment of the cortisol can be recognized more effectively. Aptamers become even closer to surface after folding, and induce more negative potential to the extended gate electrode surface. Therefore, the I_D-V_{REF} curves shift to the right direction as the concentration of the cortisol is increased (Fig. 4a). The approach of using two different buffers for incubation and measurement of the target is a normal method for electrochemical measurements^{3,4}. Even some investigations use an incubation buffer with high ionic strength and after the incubation of the target, they exploit dry environment for the measurement of the signal of the FET sensor⁵.

- 6. The schematic also included the SU-8 integrated microfluidic system but no actual attempt was made towards this integration. Figure 1c seems to be a conceptual illustration rather than a schematic that explains the operation principle of the chip that is actually fabricated and presented in this work.**

Here in the current work we have discussed the base of the sensor design, the functionalization process and the detection procedure, all being compatible with a future 3D integration perspective as reported in Fig. 1c, that is clarified to be our conceptual vision, indeed. We showed that the basic design of the proposed sensor has proper sensitivity, selectivity and stability towards the cortisol. Therefore, it can be a candidate for designing a Lab-on-Chip or Lab-on-Skin device. This conceptual figure shows how the design of the sensor can be converted to a Lab-on-Skin wearable sensory system as follows and the corresponding text is phrased as it follows:

(lines 149-188) In this study, we demonstrate a label-free cortisol detection method with an Extended Gate Field Effect Transistors (EG-FET), which is overcoming the Debye screening limitation for charge sensing by using aptamer-decorated single-layer graphene on platinum as a gate electrode. The sensing element is physically separated from the electrical transducer, enabling the possibility to implement the sensor in a 3D configuration, with a nano-MOSFET as base voltametric transducer, and the sensing electrode in the back-end-of-line (BEOL) of a CMOS process, resulting in a low power wearable sensory electronic chip. The use of atomically thin graphene is crucial to chemically bind the aptamers and bring the recognition event of the analytes within the Debye limit of detection (LOD), with high sensitivity.

The EG-FET configuration proposed and explored in this work is depicted in Fig. 1c: the gate is extended through an external electrically connected platinum electrode, which is properly functionalized with a monolayer graphene sheet, decorated with selected aptamers to address the detection of cortisol within the Debye length. The platinum/graphene electrode is then immersed in the measurement solution, where a standard Ag/AgCl reference electrode is used to electrically bias the gate of the MOSFET through the solution. Experiments have been carried out with a reference buffer and solutions with various known cortisol concentrations. Following cortisol catching, the resulting changes in the MOSFET drain current is recorded and analyzed. The significant advantage of this proposed configuration relies in using an extremely stable and reproducible standard 0.18 μ m CMOS technology node for the FET transistor as transducer, while separately functionalizing the extended electrode. This configuration results in the possibility to obtain a fully 3D lab-on-chip sensory system (Fig. 1b - bottom) with the active detecting element in the BEOL, the specific Pt/graphene/aptamer layer, and the reference electrode (e.g. another fluorinated graphene/metal electrode) in the BEOL of the CMOS process. Such a 3D chip is compatible with the recently proposed concept of Lab-On-Skin (LoS)⁸ suitable for collection and analyzing the concentrations of biomarkers

in human sweat. Such conceptual LoS system includes an SU-8 integrated microfluidic system that allows the Liquid Under Test (LUT) to flow over both the planar chlorinated reference electrode and the graphene/aptamer sheet.

- 7. Another ambiguous point is the surface modification process. The PBSE linkers are not blocked after aptamer immobilization. There is no evidence to suggest that all NHS-esters are deactivated after aptamer immobilization. Although cortisol and testosterone do not contain amine for their direct attachment to the NHS ester; tris(hydroxymethyl)aminomethane in the incubation buffer (there is a typo in SI for tris buffer) will potentially bind to electrode due to unreacted ester. Similarly, any amine-containing molecules in LUT might bind to the surface if there is excess NHS ester present. Please clarify this point.**

The target of the blocking with PBSE after aptamer functionalization is the inhibition of the attachment of the other molecules to sensor surface and prevent any unexpected signal arising from nonspecific binding of the other molecules to the electrode surface. However, there are some points:

None of the tested molecules have amine group, so they cannot attach to the probable rest of unbounded PBSE molecules. The only molecule which has the potential to attach the PBSE is tris(hydroxymethyl)aminomethane. However, if we assume that it binds to the surface, it will not be able to affect the response of the sensor, as it does not have any free charge in the pH (7.4) of the measurement solution. We investigated this assumption by examining the effect of the incubation buffer on the response of the sensor in the drift study (Supplementary information, part 4), when we switched between two buffers and recorded the response of the sensor as the I_D-V_{REF} curve. It is obvious from Fig .S6 that there is no significant shift between the I_D-V_{REF} curve of fresh functionalized electrode and the I_D-V_{REF} of the sensor, after it was kept in the incubation buffer for 30 min. Therefore, in the condition of our work. There were no logical point to use a chemical like ethanolamine (frequently used chemical for passivation of PBSE) to passivate the probable rest of the unbounded PBSE molecules. Even if we add other structurally similar hormones like Triamcinolone, Progesterone, Prednisolone, Corticosterone and Cortisone in to the measurement or incubation buffer solutions, none of them have the Amin group to be attached to surface of the electrode. If we used complex biofluid such as serum, which contain proteins, we would passivate the free PBSE molecules after aptamer functionalization. However, physical lengths of different protein complexes are

longer than the associated with physiological media^{6,9} and the response of the sensor can just be affected by the attachment of any species carrying charge within in the Debye length. Nevertheless, for fabrication of the lab on chip wearable sensory system, the passivation can improve the level of the accuracy of the sensor selectivity.

- 8. The authors should perform selectivity study with more structurally similar molecules, especially cortisone. In addition, the citation for the aptamer sequence should be provided.**

The selectivity of a sensor is the result of the specification of the sensor catch probe towards the target. In our work, the catch probe is an aptamer sequence which had been already introduced by other group³ in Scientific Reports journal for designing of an electrochemical cortisol sensor. They investigated the affinity of this aptamer sequence towards different hormones with similar structures to the cortisol, including Triamcinolone, Progesterone, Prednisolone, Cortisone and Corticosterone. The response of their sensor towards all of the tested hormones showed $\leq 5\%$ of the sensor response towards the cortisol. Especially the response of the sensor towards the cortisone was just 1% of the sensor response towards the cortisol.

We applied the testosterone, which is another adrenal hormone, for the selectivity investigation, as the selectivity of the applied aptamer sequence towards the testosterone has not yet been reported by other authors and it has been investigated as an interfering hormone in other articles related to cortisol detection^{2,4,10}.

The aptamer sequence is provided in the text of Supplementary material (note 1.4), as it follows:

Cortisol aptamer has 61 nucleotides and was synthesized by Mycosynth with 5'-amine modification. The final sequence is 5'-amine-AG CAG CAC AGA GGT CAGATG CAA ACC ACA CCT GAG TGG TTAGCG TAT GTC ATT TAC GGACC.

However, after the remark from the reviewer, we added this sequence in the caption of Fig. 2 in the main paper, as well:

(lines 224-226) (e) SEM image of the transferred graphene after functionalization with aptamers. As explained in the Supplementary Nots, the aptamer sequence used in this work is 5'-amine-AG CAG CAC AGA GGT CAGATG CAA ACC ACA CCT GAG TGG TTAGCG TAT GTC ATT TAC GGACC.

9. **Other small issues related to the manuscript are: p2 line 41: pain is a sensation not a stimulus; Labeling and formatting of Figure 4 are different from other figures.**

Thank you for your points, we fully agree. These points are edited in the new version of the article.

Reviewer 2

Authors have developed an extended-gate TFT sensor for the cortisol detection. There are many advantages with the EG and aptamer probe. However, there are several issues that should be clarified.

10. **For aptamer probe, authors have used 61-basepair aptamer, which has a reference. However, there are other sequences for the cortisol aptamer, even more different lengths of the aptamer have been reported. There are papers needed to be discussed in terms of the aptamers. (1) Dalirirad, et al, Sens. Actuat. B, 283 (2019) 79-86, (2) Jo, et al, Sens. Actuat. B, 304 (2020) 127424.**

Thank you for your useful comment.

We have clarified this point regarding the reviewer's comment in the new version of the article, also by citing the needed references, as follows:

(lines 143-148) Three different aptamers have been introduced for detection of the cortisol. they have 40¹¹, 61³ and 85^{5,12} nucleotides. The one with 85 nucleotides has been applied for a FET sensor with the detection limit of 50 nM⁵. However, for a FET sensor facing the challenge of the Debye length, the shorter length of the aptamer is expected to have better sensitivity and lower detection limit, as it has higher chance to not exceed the Debye length when it reacts with the target.

Therefore, we exploited the aptamer with 61 nucleotides to improve the sensitivity of the sensor.

11. **In the discussion of Debye length issue, there is a paper (Nakatsuka, et al, Science, 6750 (2018)) about the detection of small molecule like the cortisol. They used around 40-mers to capture the small molecules like neuro-chemicals. What will be an appropriate length for cortisol specific binding?**

On the base of the pervious explanation for question 10, the shorter sequence should provide indeed better results. Firstly, we exploited the shortest reported aptamer sequence containing 40 nucleotides for detection of the cortisol. The results of the fabricated sensor is depicted in figure below. As it can be seen, when the concentration of the cortisol is increased, almost no trend in terms of the real shift of the I_D - V_{REF} curves can be observed. The separation of the I_D - V_{REF} curves can be observed only in *strong inversion* region and these separations have no regular trend. Moreover, the separation of the I_D - V_{REF} curves is negligible in *weak inversion* region. It seems that the parameter, which affect the response of this sensor, is just the effect of change of the double layer capacitance of the gate electrode and not a modulation of the potential at surface of the extended gate by the aptamers catching cortisol. Different explanations can be identified for the non-responsivity of the shorter length, such as not sufficient resulting bending after capture to bring the relevant charge under the detection limit. In any case, we performed the study with various aptamer lengths and the optimal one by far is based on 61 nucleotide length.

Fig. 2: Response of the sensor having 40 mer aptamer as the catch probe to the different concentrations of the cortisol.

12. There are more EG TFT sensors to be compared with this study. Introduction part has missed importance of EG's roles. Authors can have more informative introduction.

The importance of the EG-FET sensors is explained in the new version of the article as follows:

(lines 105-120) Any chemical reactions at the top of the gate dielectric with the various species existing in the solution may induce a change of gate stack electrical characteristic. Therefore,

the current – voltage characteristic of the ISFET sensor can be modulated if the gate dielectric is exposed to interactions with fluids. However, in an advanced CMOS process the gate stack is part of the so called front-end-of-line (FEOL) process that is highly standardized and cannot be easily modified. To address this issue, Extended-Gate (EG) FETs have been proposed for sensing applications^{13,14}. In such sensor architecture, the base transducer is a standard nano-MOSFET while the sensing element is formed by a specific functional layer on the extension of the metal gate that can be an external electrode or a metal layer in the back-end-of-the-line (BEOL), connected to the nano-MOSFET gate. The EG-FET configuration has major advantages due to the separation of the integrated transducing element from the functional layers, including higher stability, less drift and even less temperature sensitivity¹⁵. Few research groups have attempted the design of cortisol sensors exploiting FET devices, although they have faced some challenges and, to our best knowledge, we report in this work the first EG-FET that fulfills sensitivity and selectivity performance compatible with sensing in human biofluids.

13. The level of cortisol detection was relatively high. LOD should be or could be lowered by more measurements. If the LOD cannot be improved, there should be some discussions about the reason, apart from the issues of human's cortisol levels.

The LOD is dependent on the sensitivity of the sensor. The sensitivity of the proposed sensor is limited to approximately 14.7 mV/decade. This limited sensitivity can be explained in terms of other phenomena affecting gate potential across the MOS. These phenomena include the electrical dipole χ at the interface between the metal gate and the electrolyte and the potential across the electrochemical double layer. This has been motivated already in the main text as below:

(lines 289-297) The value of χ is influenced by different microscopic phenomena such as the distribution of charges in the immobilized chemical species, and the ionic physisorption and chemisorption exchange between the modified gate and the electrolyte. As a result, the threshold voltage can be affected and hence deteriorate the sensitivity of an EG-FET¹⁶. In addition, the sensitive recognition of small molecules at low concentrations using the FET sensors may have particular challenges related to screening and size effects. Sensitive detection of small molecules at low concentrations via CNT or graphene based FET method is challenging due to the reduced electric field-effect of small size and few charge analyte and is even more difficult for uncharged analytes¹⁷.

However, we agree that this is not enough detailed and we included a new text, as per the reviewer:

(lines 334-343) As previously discussed, the sensitivity is limited by the additional phenomena affecting the χ . Moreover, it is reported that the surface of the graphene has a tendency to attract some biological molecules¹⁸. Therefore, a high concentration of the aptamer (100 μ M) is used for the functionalization of the electrode to cover the surface of the graphene by aptamers as densely as possible and minimize the free graphene spaces, and therefore decrease any unspecific attachment of the molecules on the surface of the graphene. It should be noted that a too dense population of the aptamers on the surface of the graphene may restrict them to bend freely after attachment to the cortisol as the result of space disturbance by the neighbor aptamers. This phenomenon creates a trade-off and limits the sensitivity of this sensor and the corresponding LOD.

14. There are no introduction about how the cortisol concentration of sweat and that of blood plasma serum can be correlated to be a meaningful tool as a wearable sensor.

We clarified this point in the new edited version of the article as follows:

(lines 51-53) As the secreted cortisol enters into the circulatory system, it can be found in detectable quantities in several bio-fluids in human body including saliva, sweat and urine¹⁹.

When compared to various bio-fluids, sweat is the most extensively evaluated non-invasive body fluid as it contains a lot of medical information for diagnosis. It is comparatively more convenience collect and analyze²⁰.

15. The comment about the COVID-19 does not seem to be very closely tied with cortisol sensor, especially with sweat sensor, even though the cortisol is one of glucocorticoids.

It has been recently reported that patients with COVID-19 show a marked and appropriate acute cortisol stress response in the blood serum and that this response is significantly higher in this patient cohort than in individuals without COVID-19²¹. Additionally, the levels of cortisol in patients who died of Covid-19 were significantly higher in comparison with surviving patients²². On the base of these reports, the observation of the cortisol level in human body as

one of the possible COVID-19 biomarkers appears to have a great importance. We find that citing this critical use case is relevant for the perspective of our wearable LoS system.

Reviewer 3

What are the major claims of the paper?

The authors claim this work to be the first report on a wearable electronic chip based sensor utilising an aptamer/graphene/platinum Extended Gate Field Effect transistor (EG-FET). The use of aptamers allowed the cortisol detection phenomena to take place within the Debye length of the FET, thus ensuring high sensitivity. The achieved LOD was 0.2nM and the selectivity of the system was tested using alternate molecules such as testosterone, which showed no response. Moreover, the authors claimed to have observed negligible voltage drift in the proposed system, preserving the sensitivity of the device.

16. Are they novel and will they be of interest to others in the community and the wider field?

The FET based detection of cortisol is a well-known phenomenon. However, developing selective systems is still a challenge. The authors have used an aptamer based approach in their proposed methodology, which already exists in literature but still lacks extensive study. The authors still need to highlight their advantages over some of the highly sensitive reports on similar systems (for ex. Sekar et al., *Scientific Reports*, 2019 or Jeong et al., *Biosensors and Bioelectronics*, 2019).

Thank you very much for your encouraging comments and points.

Although the mentioned (and cited in our paper) articles offer low detection limit, the maximum concentration of their linear range is limited to little amounts too. Therefore, the reported dynamic range in these papers is out of the normal cortisol concentration in sweat, saliva or urine samples.

In contrast, the reported dynamic range of our sensor is in the range of cortisol concentration in the human perspiration. In addition, the mentioned articles used antibody as the catch probe. However, we exploited aptamers, and this is explained in the main paper text as below:

(lines 138-143) Compared to antibodies, aptamers have superior advantages as catch probes as they are synthesized in vitro, reducing the batch-to-batch difference. Additionally, they can

be designed for different degrees of affinity for a targeted molecule versus a modified disturbing analog^{23,24}. Moreover, aptamers are less affected by temperature fluctuations and are more stable for long-term storage. They can be covalently immobilized on most surfaces by modifying the 5' or 3' end^{3,25}.

(lines 152-155) *The sensing element is physically separated from the electrical transducer, enabling the possibility to implement the sensor in a 3D configuration, with a nano-MOSFET as base voltametric transducer, and the sensing electrode in the back-end-of-line (BEOL) of a CMOS process, resulting in a low power wearable sensory electronic chip.*

Additionally the configuration of EG-FET offers many advantages, which is discussed in the answer of the question 12.

17. Is the work convincing, and if not, what further evidence would be required to strengthen the conclusions?

The work is convincing. The characterization and the modelling has been extensive and satisfying in justifying the results.

18. However, one small comment for the AFM study: The AFM trend needs to be explained. The SEM images seem to suggest that the roughness increases. Why is the trend different in AFM?

For SEM imaging, we used the in-lens detector, which is not able to show the topography of the surface. In addition, the contrast and brightness used during the SEM imaging could make a difference between the bare graphene monolayer and functionalized surface. However, we think that trend in the SEM images is misleading (only apparent) and cannot be objectively used for topography estimation. The only trusted data and trend for surface topography are the ones observed with the AFM images, as reported in the paper.

19. On a more subjective note, do you feel that the paper will influence thinking in the field?

Yes. As mentioned above, the authors explore using aptamers for a robust FET device. This area of research still being niche, is relevant and helpful for other researchers working in similar areas.

- 20. The authors show that $I_{DS}-V_{REF}$ curves show no significant shift at high concentrations but have a slight left shift at low concentrations. This observation has not been explained properly.**

The graph that shows the effect that is mentioned in the question is the graph of the selectivity (S6). The lowest tested concentration of the testosterone make the I_D-V_{REF} curve shift to the left direction, which is opposite to the response of the sensor to the cortisol. However, other higher concentrations of the testosterone has no significant effect on the corresponding I_D-V_{REF} curves. This phenomenon can be attributed to the some unknown chemical or physical reaction between the testosterone and this aptamer. The study of these unknown effects needs other different experimental methods such as spectroscopy and biophysical investigations, which are out of the scope of this article.

- 21. On a general note, the authors claim the biggest advantage of the system is that the folding of aptamers helps the detection to take place within the Debye length, however Debye length is also dependent on the matrix used (in this case PBS). Does it mean that the use of aptamers could be made redundant if an appropriate matrix is used? Moreover, it would be interesting to see the device performance in other matrices as well.**

Thank you for the important and useful comment. Indeed, the Debye length is dependent on the ionic strength of the matrix. We used PBS buffer, which is common physiological buffer for the bio measurements as the osmolality and ion concentrations of this solution usually match those of the human body, so we predict that the proposed sensor can work properly and with reasonable sensitivity in other matrixes with similar ion strength as the used PBS buffer. We target this study as a base for designing and fabrication of a new wearable system. However, optimization of the performance parameters of the sensor for application in complex matrix of biofluids will need additional careful analysis, which is beyond the goals of the current study.

- 22. We would also be grateful if you could comment on the appropriateness and validity of any statistical analysis, as well the ability of a researcher to reproduce the work, given the level of detail provided.**

As shown in the new figure of 4e, the results have been statistically proven repeating each measurement with the same device many times. In the new version of the article, we have included an error propagation study to document and prove this point. The new provided results

for a concentration range between 1nM-100nM (Fig. S3), demonstrate the ability of reproducing the results with a different sensors and functionalized electrodes. As required by another reviewer, we repeated a series of experiments with new devices in low concentration ranges and the results are excellently matching the initial ones. We have provided all the necessary details in the main paper and additional material to provide any other research with the ability to reproduce our results.

References:

1. Deen, M. J., Shinwari, M. W., Ranuárez, J. C. & Landheer, D. Noise considerations in field-effect biosensors. *J. Appl. Phys.* **100**, 074703 (2006).
2. Parlak, O., Keene, S. T., Marais, A., Curto, V. F. & Salleo, A. Molecularly selective nanoporous membrane-based wearable organic electrochemical device for noninvasive cortisol sensing. *Sci. Adv.* **4**, eaar2904 (2018).
3. Fernandez, R. E. *et al.* Disposable aptamer-sensor aided by magnetic nanoparticle enrichment for detection of salivary cortisol variations in obstructive sleep apnea patients. *Sci. Rep.* **7**, 1–9 (2017).
4. Sanghavi, B. J. *et al.* Aptamer-functionalized nanoparticles for surface immobilization-free electrochemical detection of cortisol in a microfluidic device. *Biosens. Bioelectron.* **78**, 244–252 (2016).
5. Xu, X. *et al.* Reconfigurable carbon nanotube multiplexed sensing devices. *Nano Lett.* **18**, 4130–4135 (2018).
6. Makowski, M. S. & Ivanisevic, A. Molecular analysis of blood with micro-/nanoscale field-effect-transistor biosensors. *Small Weinh. Bergstr. Ger.* **7**, 1863–1875 (2011).
7. Chu, C.-H. *et al.* Beyond the Debye length in high ionic strength solution: direct protein detection with field-effect transistors (FETs) in human serum. *Sci. Rep.* **7**, 1–15 (2017).

8. Bellando, F. *et al.* Lab on skinTM: 3D monolithically integrated zero-energy micro/nanofluidics and FD SOI ion sensitive FETs for wearable multi-sensing sweat applications. in *2017 IEEE International Electron Devices Meeting (IEDM)* 18.1.1-18.1.4 (2017). doi:10.1109/IEDM.2017.8268413.
9. Jang, H.-J. *et al.* Electronic cortisol detection using an antibody-embedded polymer coupled to a field-effect transistor. *ACS Appl. Mater. Interfaces* **10**, 16233–16237 (2018).
10. Wu, H., Ohnuki, H., Hibi, K., Ren, H. & Endo, H. Development of a label-free immunosensor system for detecting plasma cortisol levels in fish. *Fish Physiol. Biochem.* **42**, 19–27 (2016).
11. Dalirirad, S. & Steckl, A. J. Aptamer-based lateral flow assay for point of care cortisol detection in sweat. *Sens. Actuators B Chem.* **283**, 79–86 (2019).
12. Jo, S. *et al.* Localized surface plasmon resonance aptasensor for the highly sensitive direct detection of cortisol in human saliva. *Sens. Actuators B Chem.* **304**, 127424 (2020).
13. Khalifa, A. M., Abdulateef, S. A., Kabaa, E. A., Ahmed, N. M. & Sabah, F. A. Study of acidosis, neutral and alkalosis media effects on the behaviour of activated carbon threads decorated by zinc oxide using extended gate FET for glucose sensor application. *Mater. Sci. Semicond. Process.* **108**, 104911 (2020).
14. Palit, S. *et al.* Ultrasensitive dopamine detection of indium-zinc oxide on PET flexible based extended-gate field-effect transistor. *Sens. Actuators B Chem.* **310**, 127850 (2020).
15. Iskierko, Z. *et al.* Extended-gate field-effect transistor (EG-FET) with molecularly imprinted polymer (MIP) film for selective inosine determination. *Biosens. Bioelectron.* **74**, 526–533 (2015).
16. Estrela, P., Stewart, A. G., Yan, F. & Migliorato, P. Field effect detection of biomolecular interactions. *Electrochimica Acta* **50**, 4995–5000 (2005).

17. Tran, T.-T. & Mulchandani, A. Carbon nanotubes and graphene nano field-effect transistor-based biosensors. *TrAC Trends Anal. Chem.* **79**, 222–232 (2016).
18. Liu, B., Huang, P.-J. J., Kelly, E. Y. & Liu, J. Graphene oxide surface blocking agents can increase the DNA biosensor sensitivity. *Biotechnol. J.* **11**, 780–787 (2016).
19. Sekar, M., Pandiaraj, M., Bhansali, S., Ponpandian, N. & Viswanathan, C. Carbon fiber based electrochemical sensor for sweat cortisol measurement. *Sci. Rep.* **9**, 403 (2019).
20. Kaushik, A., Vasudev, A., Arya, S. K., Pasha, S. K. & Bhansali, S. Recent advances in cortisol sensing technologies for point-of-care application. *Biosens. Bioelectron.* **53**, 499–512 (2014).
21. Tan, T. *et al.* Association between high serum total cortisol concentrations and mortality from COVID-19. *Lancet Diabetes Endocrinol.* **8**, 659–660 (2020).
22. Ramezani, M. *et al.* The Role of Anxiety and Cortisol in Outcomes of Patients With Covid-19. *Basic Clin. Neurosci. J.* 179–184 (2020) doi:10.32598/bcn.11.covid19.1168.2.
23. Gold, L. *et al.* Aptamer-based multiplexed proteomic technology for biomarker discovery. *Nat. Preced.* 1–1 (2010) doi:10.1038/npre.2010.4538.1.
24. Baker, M. Reproducibility crisis: Blame it on the antibodies. *Nat. News* **521**, 274 (2015).
25. Song, K.-M., Lee, S. & Ban, C. Aptamers and Their Biological Applications. *Sensors* **12**, 612–631 (2012).

6th Sep 20

Dear Professor Ionescu,

Thank you for submitting your revised manuscript, "An Extended Gate Field-Effect-Transistor for Sensing the Hormone of Stress", to Communications Materials. It has now been seen again by the 3 referees, whose comments are appended below. You will see that Reviewer 2 and 3 both support publication. Reviewer 1, however, is requesting data to demonstrate the selectivity of the aptamer. We therefore invite you to revise and resubmit your manuscript, taking into account the points raised.

When submitting your revised manuscript, please include the following:

-A response letter with a point-by-point reply to each of the referee comments and a description of changes made. Please include the complete referee report in the response letter. Please note that the response letter must be separate to the cover letter to the editors.

-A marked-up version of the manuscript with all changes to the text in a different colored font. Please do not include tracked changes or comments. Please select the file type 'Revised Manuscript - Marked Up' when uploading the manuscript file to our online system.

-A clean version of the manuscript. Please select the file type 'Article File'.

-An updated <https://www.nature.com/documents/nr-editorial-policy-checklist.zip> Editorial Policy checklist, uploaded as a 'Related Manuscript File' type. This checklist is to ensure your paper complies with all relevant editorial policies. If needed, please revise your manuscript in response to these points. Please note that this form is a dynamic 'smart pdf' and must therefore be downloaded and completed in Adobe Reader. Clicking this link will download a zip file containing the pdf.

In the event that your manuscript is accepted we will provide detailed guidance on our journal policies and formatting. You may however wish to ensure that the manuscript complies with our house style at this stage. See our style and formatting guide (<https://www.nature.com/documents/commsj-phys-style-formatting-guide-accept.pdf>) and checklist (<https://www.nature.com/documents/commsj-phys-style-formatting-checklist-article.pdf>) for reference.

Data availability statements and data citations policy: All Communications Materials manuscripts must include a section titled "Data Availability" at the end of the Methods section or main text (if no Methods). More information on this policy, and a list of examples, is available at <http://www.nature.com/authors/policies/data/data-availability-statements-data-citations.pdf>.

- Accession codes for deposited data
- Other unique identifiers (such as DOIs and hyperlinks for any other datasets)
- At a minimum, a statement confirming that all relevant data are available from the authors
- If applicable, a statement regarding data available with restrictions
- If a dataset has a Digital Object Identifier (DOI) as its unique identifier, we strongly encourage including this in the Reference list and citing the dataset in the Data Availability Statement.

DATA SOURCES: We strongly encourage authors to deposit all new data associated with the paper in a persistent repository where they can be freely and enduringly accessed. We recommend submitting the data to discipline-specific, community-recognized repositories, where possible and a list of recommended repositories is provided at <http://www.nature.com/sdata/policies/repositories>.

If a community resource is unavailable, data can be submitted to generalist repositories such as [figshare](https://figshare.com/) or [Dryad Digital Repository](http://datadryad.org/). Please provide a unique identifier for the data (for example a DOI or a permanent URL) in the data availability statement, if possible. If the repository does not provide identifiers, we encourage authors to supply the search terms that will return the data. For data that have been obtained from publically available sources, please provide a URL and the specific data product name in the data availability statement. Data with a DOI should be further cited in the methods reference section.

Please use the following link to submit your documents:

[link redacted]

We hope to receive your revised paper within three months; please let us know if you aren't able to submit it within this time so that we can discuss how best to proceed. If we don't hear from you, and the revision process takes significantly longer, we will close your file. In this event, we will still be happy to reconsider your paper at a later date, as long as nothing similar has been accepted for publication at Communications Materials or published elsewhere in the meantime.

We understand that due to the current global situation, the time required for revision may be longer than usual. We would appreciate it if you could keep us informed about an estimated timescale for resubmission, to facilitate our planning. Of course, if you are unable to estimate, we are happy to accommodate necessary extensions nevertheless.

Please do not hesitate to contact me if you have any questions or would like to discuss these

revisions further. We look forward to seeing the revised manuscript and thank you for the opportunity to review your work.

Best regards,

John Plummer, PhD
Chief Editor
orcid.org/0000-0003-4824-8497
Communications Materials

Reviewers' comments:

Reviewer #1 (Remarks to the Author):

The authors addressed some of my previous comments. I think these changes greatly improved the paper. However, there are two remaining important points to be addressed before publication.

1. In the abstract, the authors claim: "capable of real-time monitoring of the circadian rhythm of cortisol in human sweat." Without data support from sweat (or other biofluids) and matrix effect study, I don't see why this device is 'capable'. The authors should not over claim.
2. In the response letter, to address my previous comment, the authors only quoted another paper saying this aptamer had good selectivity. However, this does not mean the sensor proposed in this work has similar selectivity. It is important for the authors to provide selectivity study in the paper (basically for all biosensor papers).

Reviewer #2 (Remarks to the Author):

- Since discussion issues raised from reviewer were largely resolved, it can be acceptable.

Reviewer #3 (Remarks to the Author):

The authors have not only answered my queries very satisfactorily and convincingly, but in my opinion they have also answered the other reviewer's queries well. I have no further queries for them.

Reviewer 1

The authors addressed some of my previous comments. I think these changes greatly improved the paper. However, there are two remaining important points to be addressed before publication.

1. In the abstract, the authors claim: "capable of real-time monitoring of the circadian rhythm of cortisol in human sweat." Without data support from sweat (or other biofluids) and matrix effect study, I don't see why this device is 'capable'. The authors should not over claim.

Answer:

The reviewer remark is considered constructively, as we are not reporting here in-vivo experiments but important validations steps towards such goal. On the other hand, we have: (i) carefully selected ranges of concentrations in all our experiments that fully correspond to the ranges of concentrations in human sweat, (ii) carefully checked the cross-sensitivity to the main two other hormones (testosterone and cortisone) and demonstrated that our sensor and functionalization are highly selective. To follow the reviewer remark, we have changed our phrasing for the respective claim. The article mentions later that we do not report validations in the complex matrix of sweat but we consider all our data as very significant and promising steps towards such capability.

'In this work we report the first wearable sensory electronic chip using a label-free detection based on a platinum/graphene aptamer Extended Gate Field Effect transistor (EG-FET) embodiment for the recognition of cortisol in biological buffers, showing promising experimental features for future real-time monitoring of the circadian rhythm of cortisol in human sweat. This sensor is an excellent candidate for integrated miniaturized lab-on-chip and/or lab-on-skin wearable sensory systems.

... The new EG-FET achieves a detection limit near 0.2 nM over a wide range, between 1 nM and 10 μM, of cortisol concentrations in a physiological fluid, with negligible drift over time and high selectivity. The range of tested concentrations fully covers the ones in the sweat and the selectivity is tested in vitro versus cortisone and testosterone.'

2. In the response letter, to address my previous comment, the authors only quoted another paper saying this aptamer had good selectivity. However, this does not mean the sensor proposed in this work has similar selectivity. It is important for the authors to provide selectivity study in the paper (basically for all biosensor papers).

Answer:

We have fabricated new functionalized electrodes and carried a new experiment to clarify this particular aspect of the article on which the reviewer insisted.

As it can be observed in the figure presented in this section of the answer, for such delicate investigation of the sensor selectivity, we did a new experiment and tested the sensor response towards the Cortisone, which is the metabolized form of the cortisol in the peripheral tissues and has the **most similar structure** to the cortisol among other hormones¹.

The results of this investigation are depicted in the new fig. S6b of the supplementary information and are shown here too.

Response of the sensor towards different concentrations of cortisone as an interfering biomolecules.

As can be seen from the above figure, when the sensor is exposed to the lowest concentration of the cortisone (1 nM), the corresponding $I_{DS}-(V_{REF}-V_T)$ curve shifts a little to the right direction compared to the characteristic of the sensor in a buffer with no cortisone. However, as the cortisone

concentration gradually increases up to 10 μM , the $I_{\text{DS}}\text{-}V_{\text{REF}}$ curves overlap and no trend was observed.

These results are in strong contrast to the clear and strong sensor response to cortisol (for which all curves are shifting systematically to the right when the cortisol concentration increases). In addition, the shift of the $I_{\text{DS}}\text{-}V_{\text{REF}}$ characteristics in terms of V_{REF} voltage at constant current of 1 μA was measured under the presence of cortisol alone and under the combination of both cortisol and cortisone. When the highest concentration of the cortisone (10 μM) was added to the solution containing lowest tested concentration of the cortisol, almost no shift of $I_{\text{DS}}\text{-}V_{\text{REF}}$ curve was observed.

Therefore, it is concluded that our sensor is highly selective to cortisol. We hope that the reviewer will be fully satisfied with this new experiments that involved a significant additional effort.

References

1. Rippe, R. C. A. *et al.* Splitting hair for cortisol? Associations of socio-economic status, ethnicity, hair color, gender and other child characteristics with hair cortisol and cortisone. *Psychoneuroendocrinology* **66**, 56–64 (2016).

Notes:

- 1) We have also corrected a typo error in the main paper in equation (1) where V_{ref} is replaced by E_{ref} .
- 2) Highlights - All the corrections of *Revision 1* are highlighted in yellow and all the new changes done in *Revision 2* are highlighted in green.

30th Oct 20

Dear Professor Ionescu,

Your manuscript titled "An Extended Gate Field-Effect-Transistor for Sensing the Hormone of Stress" has now been seen again by Reviewer 1, whose comments appear below. In light of their advice I am delighted to say that we are happy, in principle, to publish a suitably revised version in Communications Materials under the open access CC BY license (Creative Commons Attribution v4.0 International License).

We therefore invite you to edit your manuscript to comply with our journal policies and formatting style in order to maximise the accessibility and therefore the impact of your work.

EDITORIAL REQUESTS

* Your manuscript should comply with our policies and format requirements, detailed in our style and formatting guide (<https://www.nature.com/documents/commsj-phys-style-formatting-guide-accept.pdf>).

* Please edit your manuscript according to the editorial requests in the attached table, and outline revisions made in the right hand column. If you have any questions or concerns about any of our requests, please do not hesitate to contact me. It is important that each request be addressed in order to avoid delays in accepting your manuscript. Please upload the completed table with your manuscript files.

* The editorial requests table also includes a full list of the files that must be provided upon resubmission. Please upload your files according to this table.

* An updated editorial policy checklist that verifies compliance with all required editorial policies must be completed and uploaded with the revised manuscript. All points on the policy checklist must be addressed; if needed, please revise your manuscript in response to these points. Please note that this form is a dynamic 'smart pdf' and must therefore be downloaded and completed in Adobe Reader. Clicking this link will download a zip file containing the pdf.

OPEN ACCESS

Communications Materials is a fully open access journal. Articles are made freely accessible on publication under a [CC BY](http://creativecommons.org/licenses/by/4.0) license (Creative Commons Attribution 4.0 International License). This license allows maximum dissemination and re-use of open access materials and is preferred by many research funding bodies.

For further information about article processing charges, open access funding, and advice and support from Nature Research, please visit <https://www.nature.com/commsmat/about/open-access>

RESUBMISSION

At acceptance, the corresponding author will be required to complete an Open Access Licence to Publish on behalf of all authors, declare that all required third party permissions have been obtained and provide billing information in order to pay the article-processing charge (APC) via credit card or invoice.

Please note that your paper cannot be sent for typesetting to our production team until we have received these pieces of information; therefore, please ensure that you have this information ready when submitting the final version of your manuscript.

Please use the following link to submit your revised files:

[link redacted]

We hope to hear from you within two weeks; please let us know if the process may take longer.

Best regards,

John Plummer, PhD
Chief Editor
orcid.org/0000-0003-4824-8497
Communications Materials

REVIEWERS' COMMENTS:

Reviewer #1 (Remarks to the Author):

The authors have addressed my remaining comments. Now the paper is suitable for publication.